# Identifying Molecular Signatures of Distinct Modes of Collective Migration in Response to the Microenvironment Using Three-Dimensional Breast Cancer Models

**DOI:** 10.3390/cancers13061429

**Published:** 2021-03-20

**Authors:** Diana Catalina Ardila, Vaishali Aggarwal, Manjulata Singh, Ansuman Chattopadhyay, Srilakshmi Chaparala, Shilpa Sant

**Affiliations:** 1Department of Pharmaceutical Sciences, School of Pharmacy, University of Pittsburgh, Pittsburgh, PA 15261, USA; dca18@pitt.edu (D.C.A.); VAA30@pitt.edu (V.A.); Manjulata.Singh@vcuhealth.org (M.S.); 2Health Sciences Library System, University of Pittsburgh, Pittsburgh, PA 15219, USA; ansuman@pitt.edu (A.C.); srichaparala@pitt.edu (S.C.); 3Department of Bioengineering, Swanson School of Engineering, University of Pittsburgh, Pittsburgh, PA 15219, USA; 4McGowan Institute for Regenerative Medicine, University of Pittsburgh, Pittsburgh, PA 15260, USA; 5UPMC-Hillman Cancer Center, Pittsburgh, PA 15260, USA

**Keywords:** tumor microenvironment, tumor-intrinsic hypoxia, epithelial-mesenchymal transition (EMT), collective migration, three-dimensional cultures, microtumors, microarray, bioinformatic analysis

## Abstract

**Simple Summary:**

The objective of this study was to investigate the role of two microenvironmental factors, namely, tumor-intrinsic hypoxia and secretome in inducing collective migration. We utilized three-dimensional (3D) discrete-sized microtumor models, which recapitulate hallmarks of transition of ductal carcinoma in situ (DCIS) to invasive ductal carcinoma (IDC). Tumor-intrinsic hypoxia induced directional migration in large hypoxic microtumors while secretome from large microtumors induced radial migration in non-hypoxic microtumors. This highlights the emergence phenotypic heterogeneity and plasticity in cancer cells in response to different microenvironmental stimuli. To unravel mechanisms underlying these two distinct modes of migration, we performed differential gene expression analysis of hypoxia- and secretome-induced migratory phenotypes using non-migratory, non-hypoxic microtumors as controls. We proposed unique gene signature sets related to tumor-intrinsic hypoxia, hypoxia-induced epithelial-mesenchymal transition (EMT), as well as hypoxia-induced directional migration and secretome-induced radial migration.

**Abstract:**

Collective cell migration is a key feature of transition of ductal carcinoma in situ (DCIS) to invasive ductal carcinoma (IDC) among many other cancers, yet the microenvironmental factors and underlying mechanisms that trigger collective migration remain poorly understood. Here, we investigated two microenvironmental factors, tumor-intrinsic hypoxia and tumor-secreted factors (secretome), as triggers of collective migration using three-dimensional (3D) discrete-sized microtumor models that recapitulate hallmarks of DCIS-IDC transition. Interestingly, the two factors induced two distinct modes of collective migration: directional and radial migration in the 3D microtumors generated from the same breast cancer cell line model, T47D. Without external stimulus, large (600 µm) T47D microtumors exhibited tumor-intrinsic hypoxia and directional migration, while small (150 µm), non-hypoxic microtumors exhibited radial migration only when exposed to the secretome of large microtumors. To investigate the mechanisms underlying hypoxia- and secretome-induced directional vs. radial migration modes, we performed differential gene expression analysis of hypoxia- and secretome-induced migratory microtumors compared with non-hypoxic, non-migratory small microtumors as controls. We propose unique gene signature sets related to tumor-intrinsic hypoxia, hypoxia-induced epithelial-mesenchymal transition (EMT), as well as hypoxia-induced directional migration and secretome-induced radial migration. Gene Set Enrichment Analysis (GSEA) and protein-protein interaction (PPI) network analysis revealed enrichment and potential interaction between hypoxia, EMT, and migration gene signatures for the hypoxia-induced directional migration. In contrast, hypoxia and EMT were not enriched in the secretome-induced radial migration, suggesting that complete EMT may not be required for radial migration. Survival analysis identified unique genes associated with low survival rate and poor prognosis in TCGA-breast invasive carcinoma dataset from our tumor-intrinsic hypoxia gene signature (CXCR4, FOXO3, LDH, NDRG1), hypoxia-induced EMT gene signature (EFEMP2, MGP), and directional migration gene signature (MAP3K3, PI3K3R3). NOS3 was common between hypoxia and migration gene signature. Survival analysis from secretome-induced radial migration identified ATM, KCNMA1 (hypoxia gene signature), and KLF4, IFITM1, EFNA1, TGFBR1 (migration gene signature) to be associated with poor survival rate. In conclusion, our unique 3D cultures with controlled microenvironments respond to different microenvironmental factors, tumor-intrinsic hypoxia, and secretome by adopting distinct collective migration modes and their gene expression analysis highlights the phenotypic heterogeneity and plasticity of epithelial cancer cells.

## 1. Introduction

Approximately 1 in 5 of breast cancers detected through mammography are pre-invasive ductal carcinoma in situ (DCIS) [1]. If left untreated, DCIS will progress to more deadly invasive ductal carcinoma (IDC) [2]. Lack of mechanistic understanding on how a pre-malignant breast cancer in situ develops into a malignant invasive breast cancer contributes to a growing problem of inadequate clinical treatment. The existing evidence that pre-invasive DCIS and IDC exhibit comparable genomic profiles [3,4] suggests that the transition to an invasive breast cancer is driven by microenvironmental factors prevalent in the pre-malignant phenotype, and not by genetic abnormalities in DCIS. Tumor-intrinsic hypoxia is one of the hallmarks of DCIS microenvironment, and is associated with phenotypic changes that may lead to a more aggressive behavior [5].

Recently, IDC was shown to exhibit multicellular cohesive invasion, in which the cells migrate collectively to invade the surrounding extracellular matrix (ECM) [6]. Such collective migratory behavior is also observed in other aggressive cancers and has been associated with invasion into surrounding tissue and distant metastasis [7,8]. Given that tumor invasion and metastasis are the leading causes of cancer mortality [9], it is important to understand the emergence of migratory phenotypes that contribute to poor clinical outcomes. Collective migration in IDC is attributed to the conserved expression of E-cadherin and other cell-cell junction molecules, along with the upregulation of mesenchymal markers observed in human tissue samples and mouse models of IDC [8]. Collectively migrating cell groups exhibit different movement dynamics, which depend on the cell-ECM interactions, cell adhesion systems, and the status of epithelial and mesenchymal markers in the tumor cell population, all of which are influenced by the crosstalk between tumor cells and surrounding microenvironment [10,11].

The tumor microenvironment plays an important role in emergence of aggressive phenotypes [11]. Solid tumors including DCIS develop hypoxic microenvironment in the core of the tumor away from the blood supply (hereafter tumor-intrinsic hypoxia) due to lack of oxygen and nutrient supply [12,13]. Such hypoxic microenvironments can activate intrinsic tumor signaling, which further initiates migratory events in cancer cells [14,15,16,17,18]. Hypoxia can also induce tumor-secreted factors (secretome) that modify the tumor microenvironment and act as extrinsic signaling to trigger a migratory and invasive behavior [16,17,19]. Diversity reported in migration modes and mechanisms of collective migration are dictated by the local tumor microenvironment [11,20]. However, microenvironmental factors and underlying mechanisms that trigger collective migration remain poorly understood.

Given that microenvironmental factors such as hypoxia and secretome influence the migratory behavior of cancer cells, it is important to develop an experimental system that can reproducibly recapitulate the spontaneous emergence of tumor-intrinsic hypoxic microenvironment along with tumor-secreted factors. Recently, different three-dimensional (3D) in vitro models such as organoids, scaffold-based, and microfluidic models are being exploited to study the effect hypoxic microenvironment on tumor cells [21]. These models aim to recreate the hypoxic environment by external hypoxia using hypoxic chambers [22,23], scaffolds with different thicknesses or densities to create oxygen gradients [24,25], or by using external stimulus such as human stromal-cell derived factor-1 (SDF-1) gradient [26,27]. Culture of cancer cells in hypoxic chambers under reduced oxygen tension cannot reproduce the spatial oxygen gradients and spatial heterogeneity observed in solid tumors in vivo, where only an inner hypoxic core exists. This is because, in hypoxic chambers, all cells irrespective of their position are exposed to hypoxia, unlike what is observed in vivo. Exposure to external stimulus like chemokines/cytokines although recreates chemotaxis, it does not necessarily mimic global hypoxic environments. Thus, these approaches fail to recreate the naturally formed, dynamic, tumor-intrinsic hypoxic microenvironment observed in solid tumors.

To overcome this challenge, we engineered size-controlled 3D microtumor models in which the microenvironment is solely defined by the microtumor size and tumor-secreted factors [16,17,28,29,30]. In this system, we can recapitulate two different modes of collective migration in response to two different microenvironmental factors: Tumor-intrinsic hypoxia and secretome. This is achieved by reproducibly generating hundreds of uniform size, small (150 µm) microtumors that are non-hypoxic and non-migratory, and large (600 µm) microtumors that spontaneously develop tumor-intrinsic hypoxia without any external stimulus, and exhibit directional migration [16,17,29]. The directional migratory phenotype induced by tumor-intrinsic hypoxia was irreversible as cells from directionally migrating tumors remained migratory even after being dissociated and regrown as non-hypoxic small microtumors [16]. Notably, although hypoxia triggers directional migration, it does not sustain the migratory phenotype since inhibition of hypoxia inducible factor 1 subunit alpha (HIF1a) is effective in preventing migration only when the treatment is started at early stages, and fail to prevent migration when the treatment is started at later stages [17]. We have also shown that the conditioned media (CM, secretome) of the large directionally migrating tumors contain elevated levels of soluble E-cadherin (sE-CAD), Fibronectin (Fib), and Matrix metalloproteinase-9 (MMP9) and induced migratory behavior in small non-hypoxic and non-migratory microtumors [17]. Treatment with sE-CAD alone could induce the migratory phenotype in non-migratory 150 µm microtumors. [17]. It should be noted that both, the large directional and small radial migrating tumors moved collectively and conserved the cell-cell junction markers E-cadherin (E-CAD) during the entire migratory process [17], which is a key characteristic of collective migration [31,32].

The goal of this study is to investigate mRNA expression changes induced by tumor-intrinsic hypoxia and secretome as two microenvironmental factors and elucidate the plausible downstream mechanisms responsible for emergence of two distinct directional and radial migratory phenotypes using our microtumor models with controlled microenvironments. Here, we analyzed the changes in gene expression profiles obtained by microarray of these migratory microtumors compared to non-hypoxic and non-migratory microtumors. In our approach, we identified differentially expressed genes (DEGs) in tumor intrinsic hypoxia-induced directional migration (hereafter directional migration) vs. non-hypoxic small tumors, and secretome-induced radial migration (hereafter radial migration) also compared to non-hypoxic small tumors. To uncover the molecular mechanisms using DEGs, we performed bioinformatics analysis using multiple enrichment analysis tools and identified enriched gene ontology (GO) terms and biological pathways. We then constructed unique gene expression signature profiles for tumor-intrinsic hypoxia, hypoxia-induced epithelial-mesenchymal transition (EMT), hypoxia-induced directional migration, and secretome-induced radial migration, and evaluated the interaction between them. We performed survival analysis to evaluate the usefulness of the signature profiles to be used as a prognosis tool.

## 2. Material and Methods

### 2.1. Cell Lines and Cell Culture

The breast cancer cell line (T47D) was purchased from American type culture collection (ATCC). Cell culture supplies and media were obtained from Corning^®^ and Mediatech^®^, respectively, unless specified. The cells were passaged and maintained in T75 flasks in Dulbecco’s modified Eagle’s medium (DMEM) (MT10013CV, Corning^®^, Lawrenceville, VA, USA) supplemented with 10% fetal bovine serum (FBS) (S11250, Atlanta biologicals, Flower Branch, GA, USA), and 1% penicillin-streptomycin (300-002-CI, Corning^®^, Christiansburg, VA, USA) in a humidified atmosphere at 37 °C and 5% CO_2_. Cells were maintained below 60% confluency for further seeding into hydrogel microwell devices. Cell line authentication was done by University of Arizona, Genetics Core by using PowerPlex16HS PCR Kit as previously described [17].

### 2.2. Microtumor Fabrication

Microtumors of 150 µm (referred to as ‘small’ microtumors) and 600 µm diameters (referred to as ‘large’ microtumors) were obtained by seeding T47D cells in non-adhesive polyethylene glycol (PEG) hydrogel microwell devices, as previously described [16,17,29,33]. Briefly, 1 × 1 cm^2^ microwell devices were fabricated using polydimethyl siloxane (PDMS) stamps containing posts of either 150 µm or 600 µm in diameter with equal height. A solution of 20% *w*/*v* polyethylene glycol dimethacrylate (PEGDMA, 1000Da, Polysciences, Inc., USA) containing photoinitiator (Irgacure-1959, 1% *w*/*v*, Ciba AG CH-4002, Basel, Switzerland) was crosslinked under the PDMS stamps using OmniCure S2000 curing station (200W Lamp, 5 W/cm^2^, EXFO, Mississauga, ON, Canada). The microwell hydrogel devices were sterilized by submerging in 70% ethanol under UV light for 1 h under laminar air flow. Sterilized devices were then washed three times with Dulbecco’s phosphate buffered saline (DPBS) without calcium and magnesium (Corning™, USA, catalog #21-031-CV). For cell seeding, a suspension of 1.0 × 10^6^ T47D cells (less than 15 passages) in 50 µL of growth media was dropped on each hydrogel microwell device and cells were allowed to settle in the microwells for 15–30 min and those outside the microwells were removed by gentle washing with DPBS. The cell-seeded devices were cultured at 37 °C and 5% CO_2_ in a humidified incubator. To study tumor migration due to tumor-intrinsic hypoxia, 600 µm microtumors were cultured for up to 6 days (referred to as 600D6 hereafter) with replacement of 50% media with equal quantity of fresh media every day. To study secretome-induced migration, 150 µm microtumors were treated with conditioned media of 600 µm microtumors (denoted as ‘600/CM’) starting from day 3 to day 6 (referred to as 150CM hereafter), with replacement of 50% media with equal amount of 600/CM every day as described earlier [17]. Small 150 µm microtumors that are shown to be non-hypoxic and non-migratory were cultured similarly for 6 days with 50% fresh media change every day (referred to as 150D6 hereafter) and used as non-hypoxic and non-migratory controls.

### 2.3. Microarrays and Bioinformatic Analysis

#### 2.3.1. Microarrays

At the end of 6-day culture, RNA was isolated from 150D6, 600D6, and 150CM microtumors using GenElute™ Mammalian Total RNA Miniprep Kit (Sigma-Aldrich, St. Louis, MO, USA, cat #RTN70-1KT) and DNA cleanup was done by On-Column DNase I Digestion Set (Sigma-Aldrich, St. Louis, MO, Cat#DNASE70-1SET) as per manufacturer’s protocols. RNA quantity was measured by absorbance ratio at 260/280 nm and integrity was verified on 1% agarose gel electrophoresis. cDNA preparation, hybridization to GeneChips, scanning, and first-level data analysis were performed at the University of Pittsburgh HSCRF Genomics Research Core as follows: Biotinylated cDNA was prepared according to the standard Affymetrix Pico protocol from 5 ng total RNA (GeneChip Pico Reagent Kit User Guide, Rev. 4). Following fragmentation and labeling, 2.4 µg of cDNA were hybridized for 16 h at 45 °C with 60 RPM rotation on GeneChip Clariom_S_Human Genome Array. GeneChips were washed and stained on an Affymetrix Fluidics Station 450. GeneChips were scanned using a GeneChip 3000 scanner with 7G upgrade and autoloader. First level data analysis will be performed using Affymetrix Expression Console 1.2.0.20 using RMA_gene_full_signal normalization algorithm.

#### 2.3.2. Bioinformatic Analysis

Microarray data is deposited in the Gene Expression Omnibus database at the National Center for Biotechnology Information (GSE166211). The workflow of our bioinformatic analysis is outlined in Scheme 1 and is described in detail as follows: Differential gene expression in migratory 600D6 induced by tumor-intrinsic hypoxia or in 150CM induced by secretome was analyzed by normalization with the 150D6 (groups 600D6 vs. 150D6 and 150CM vs. 150D6, respectively, Scheme 1).

Identified differentially expressed genes (DEGs) in tumor-intrinsic hypoxia-induced directional migration (600D6 vs. 150D6) and in secretome-induced radial migration (150CM vs. 150D6) using TACIdentify statistically enriched Gene ontology biological processes and Hallmarks associated with tumor-intrinsic hypoxia-induced directional migration and secretome-induced radial migration using GSEAIdentify statistically enriched Gene ontology biological processes associated with tumor-intrinsic hypoxia-induced directional migration and secretome-induced radial migration using Correlation EngineIdentify statistically enriched pathways associated with tumor-intrinsic hypoxia-induced directional migration and secretome-induced radial migration using Ingenuity Pathway AnalysisExamine the enrichment of Hallmark hypoxia, GO EMT, and GO Tissue migration in the DEGs of tumor-intrinsic hypoxia-induced directional migration and in secretome-induced radial migration using GSEAExamine the significance of correlation of DEGs in tumor-intrinsic hypoxia-induced directional migration and secretome-induced radial migration with Go response to hypoxia and GO regulation of cell migration using BaseSpace Correlation EngineCompilation of resultant genes from GSEA and BaseSpace Correlation Engine to obtain signature gene sets associated with tumor-intrinsic hypoxia, hypoxia-induced EMT, hypoxia-induced directional migration, and secretome-induced radial migrationAnalysis of protein-protein interactions of gene signature sets in directional migration (tumor-intrinsic hypoxia, hypoxia-induced EMT, hypoxia-induced directional migration signature gene sets) and secretome-induced radial migration using NetworkAnalystIdentify statistically enriched pathways associated with the signature gene sets using KEGGSurvival analysis of genes in signature gene sets using SurvExpress and the human protein atlas

#### 2.3.3. Identification of Differentially Expressed Genes

Transcriptome Analysis Console (TAC 4.0, ThermoFisher Scientific, Waltham, MA, USA) was used to perform statistical analysis on each comparison using one-way ANOVA. The initial assessment of the microarray expression data was performed by hierarchical clustering using a cut-off *p*-value of 0.01. All the genes with False Discovery Rate (FDR) ≤ 0.05, *p*-value ≤ 0.05 and fold change ± 2 were considered significantly different in terms of gene expression and were selected for further analysis (Step 1). Volcano plots highlighting differentially expressed genes (DEGs) were reconstructed using Bioconductor v3.9 in R [34].

#### 2.3.4. Statistical Enrichment Analysis

A global analysis of the DEGs was carried out by gene ontology enrichment, hallmark enrichment, and pathway enrichment analysis. Gene set enrichment analysis (GSEA) of gene ontology and hallmarks was performed using the GSEA software (UC San Diego and Broad Institute, San Diego, CA, USA) (Step 2). The gene ontology gene sets in GSEA are part of the molecular signatures database (MSigDB) collections and contain 14750 gene sets containing genes annotated by the same ontology term. This gene ontology sets are divided into three components: biological processes (BP), cellular component (CC), or molecular function (MF). Here, we focused on BP gene sets. The gene functional classification of BP was carried out using DAVID bioinformatic database (DAVID Bioinformatics resources 6.8, NIAID/NIH) (https://david.ncifcrf.gov/ accessed on 3 March 2021). Hallmark gene sets are also part of MSigDB collections and comprise 50 gene sets that represent well-defined biological processes. Gene ontology BP and Hallmark gene set enrichment (minimum size cut off = 5) were carried out using a pre-ranked list based on fold change mapped with the MSigDB human symbol v7.1 platform [35,36,37].

To perform a more comprehensive statistical enrichment, we repeated the gene ontology enrichment of BP using BaseSpace Correlation Engine (CE) (Illumina v2.0, USA) (Step 3).

Pathway enrichment analysis was performed using the Ingenuity Pathway Analysis (IPA) suite of tools v01-16 (QIAGEN Digital Insights, Redwood City, CA, USA) to identify the canonical pathways that are enriched and differentially regulated due to tumor-intrinsic hypoxia and secretome as microenvironmental factors inducing collective migration (Step 4). The list of DEGs between migratory 600D6 induced by tumor-intrinsic hypoxia or in 150CM induced by secretome vs. non-hypoxic, non-migratory tumors was uploaded, and the core analysis was performed in IPA using Clariom S human array as a reference set. Significant differentially regulated canonical pathways were identified with a threshold −log10 (*p*-value) of >1.3 using right-tailed Fisher Exact Probability Tests. By calculating the z-scores, we determined the activity status of each pathway. The z-score gives a statistical measurement of the gene expression pattern in the dataset for each pathway compared to the expected pattern of expression based on literature [38].

A focused analysis was performed to investigate the role of genes related to tumor intrinsic hypoxia, EMT, and migration in the directional and radial migration patterns. GSEA was used to obtain the enrichment score of ‘hallmark hypoxia’, and the gene ontology terms ‘epithelial to mesenchymal transition’, and ‘tissue migration’ in DEGs from directional migratory tumors, and radial migratory tumors (Step 5). CE was used to analyze the significance of association of DEGs from directional and radial migration groups with BP such as ‘Response to hypoxia’ and ‘Regulation of cell migration’ from MSigDB.

#### 2.3.5. Meta-Analysis

A meta-analysis was carried out to validate our results with the ones in other studies that correlated genomic changes with hypoxic events. For this purpose, we used the meta-analysis tool in CE to correlate the highly regulated genes in our study groups with the ones found in seven publicly available studies (GSE3893 [39], GSE19123 [40], GSE29406 [41], GSE70805 [42,43], GSE47533 [44], GSE9649 [45,46], and GSE30019 [47]) that studied the effect of hypoxia in the gene expression of breast cancer cell lines or breast cancer tissue (Appendix A) (Step 6).

#### 2.3.6. Signature Gene Sets

Hypoxia, EMT, and migration-related genes from the previous GSEA and CE analysis were extracted and grouped in unique signature gene sets, namely ‘Tumor-intrinsic hypoxia’, ‘Hypoxia-induced EMT’, ‘Hypoxia-induced directional migration’, and ‘Secretome-induced radial migration’ signature gene sets (Step 7).

#### 2.3.7. Protein-Protein Interaction Networks

Signature gene sets related to each directional and radial migratory tumors were combined and used to generate the minimum protein-protein interaction (PPIs) networks based on the InnateDB Interactome [48] using NetworkAnalyst v3.0 [49] (Step 8). Visualization of PPIs was done in Cytoscape v3.7.2 [50]. One of the features of NetworkAnalyst is the possibility to perform functional analysis of the constructed PPI network. Then the Kyoto Encyclopedia of Genes and Genomes (KEGG) pathway enrichment was done using the genes participating in the minimum PPI network directly in NetworkAnalyst using a statistical significance of *p*-value < 0.05 (Step 9).

#### 2.3.8. Survival Analysis

Survival analysis of signature genes related to tumor-intrinsic hypoxia, hypoxia-induced EMT, as well as hypoxia-induced directional migration and secretome-induced radial migration (*n* = 174) was carried out using The Cancer Genome Atlas (TCGA) [51] breast invasive carcinoma dataset in SurvExpress [52,53]. All 174 genes were analyzed for Cox survival analysis using ‘breast’ tissue, and ‘BRCA-TCGA’ breast invasive carcinoma dataset with 962 patients and censored for survival days in the software. Similarly, protein expression analysis was performed using The Human Protein Atlas (HPA) [54,55,56] for hypoxia, EMT, and directional and radial migration signature genes, which were able to significantly predict survival in invasive breast cancer patients. Corresponding protein expression analysis by immunohistochemistry was presented for normal and breast cancer tissues (Step 10).

## 3. Results

### 3.1. Three-Dimensional Microtumor Models Exhibit Two Distinct Modes of Collective Migration in Response to Different Microenvironmental Factors: Tumor-Intrinsic Hypoxia and Secretome

We engineered discrete-sized 3D microtumor models to recapitulate DCIS-IDC transition [16,17,29], which exhibit distinct modes of collective migration, namely, directional and radial migration in response to two different microenvironmental stimuli, tumor-intrinsic hypoxia and tumor-secreted factors or ‘secretome’, respectively (Figure 1). In our system, we can reproducibly generate two distinct phenotypes from the same non-invasive parent T47D breast cancer cells, namely, small (150 µm) non-hypoxic, non-migratory microtumors, and large (600 µm) migratory microtumors that develop intrinsic hypoxia (Figure 1A,B). Our results demonstrate that large hypoxic microtumors exhibit directional collective migration starting from day 3 in culture with almost total migration of the tumors outside the microwells by day 6 (600D6). In contrast, small microtumors remain non-migratory in the microwells from day 1 to day 6 (150D6) (Figure 1B). When treated with conditioned media (CM) from large hypoxic (600 µm) microtumors, small microtumors (150CM) exhibit non-directional, radial collective migration (Figure 1C). Here, we analyzed the changes in gene expression obtained by microarray to understand the genomic differences in spontaneous emergence of these two distinct, directional and radial migratory phenotypes induced in 3D T47D microtumors by tumor-intrinsic hypoxia (600D6) and secretome (150CM), respectively. We used non-hypoxic, non-migratory microtumors (150D6) as controls (Figure 1D).

### 3.2. Global Changes in Gene Expression Induced by Tumor-Intrinsic Hypoxia and Secretome

#### 3.2.1. Differentially Expressed Genes

Hierarchical clustering of gene expression data showed that independent biological replicates for each of the microtumor groups (600D6, 150CM, and 150D6) clustered together, demonstrating the reliability of our 3D microwell system to generate independent and consistent gene expression profiles. Experimental groups with similar size microtumors, namely 150D6 and 150CM, clustered in the first level followed by large microtumors (600D6), indicating more changes in gene expression of 600D6 induced by tumor-intrinsic hypoxia compared to those induced by secretome in 150CM microtumors (Appendix A). Using a comparative gene expression analysis, we identified 1992 differentially expressed genes (DEGs) in the directionally migrating microtumors (600D6 vs. 150D6) induced by tumor-intrinsic hypoxia, which include 734 upregulated genes and 1258 downregulated genes. We also identified 305 DEGs in the radially migrating microtumors (150CM vs. 150D6) induced by secretome, of which 189 genes were upregulated and 116 genes were downregulated. Volcano plots of genes based on intensity values showed that tumor-intrinsic hypoxia had a greater effect on modulating gene expression compared to the secretome (Figure 2A).

#### 3.2.2. Gene Ontology and Pathway Enrichment Analysis

We performed a comprehensive statistical enrichment of GO: BP in directional migration and radial migration (Figure 2B). GSEA GO terms that showed opposite direction of enrichment mostly coded for the cell-cell interaction, ECM components, cytoskeletal and actin filament organization, tumor-secreted factors and response to stress. In tumor-intrinsic hypoxia (600D6); ‘positive regulation of locomotion’ was the most enriched biological process followed by ‘sensory perception’, ‘cell activation’, ‘cell-cell adhesion’, and ‘cellular response to toxic substance’ (Figure 2B(i)). While these adhesion-related processes were upregulated in hypoxic directional migration, they were downregulated in secretome-induced radial migration, highlighting role of cell adhesion and cell junction organization in hypoxia-induced directional migration in large microtumors. In the secretome-induced radial migration (150CM), the GO terms related to cytoskeletal and actin filament organization, response to type-1 interferon were enriched and upregulated (Figure 2B(ii)) in contrast to the hypoxia-induced directional migration, suggesting significance of cytoskeletal organization in radial migration.

Using GSEA hallmark analysis, we identified upregulated and downregulated hallmark gene sets (Figure 2C) enriched in the hypoxia-induced and secretome-induced migratory microtumors. We identified 21 upregulated and 15 downregulated hallmarks in the hypoxia-induced directional migration. In this group, the top upregulated hallmark gene set was ‘Hypoxia’ while the top downregulated gene set was ‘G2M checkpoint’. In secretome-induced radial migration, only two hallmark gene sets were enriched and upregulated, namely, ‘TNFα signaling via NF-κB’, and ‘mTORC1 signaling’.

The ingenuity pathway analysis (IPA) software was used to perform statistical enrichment analysis of molecular pathways. A total of 111 canonical pathways were upregulated while 204 pathways were downregulated in hypoxia-induced directional migration. Of these, ‘sirtuin signaling pathway’, ‘cell cycle: G2/M DNA damage checkpoint regulation’, ‘senescence pathway’, ‘unfolded protein response’, and ‘p53 signaling’ were top five upregulated pathways, whereas ‘cell cycle control of chromosomal replication’, ‘superpathway of cholesterol biosynthesis’, ‘aryl hydrocarbon receptor signaling’, ‘cholesterol biosynthesis I’, and ‘cholesterol biosynthesis II’ were top five downregulated pathways (Appendix A). In secretome-induced radial migration, 49 pathways were upregulated, and 23 pathways were downregulated. Of these, ‘unfolded protein response’, ‘interferon signaling’, ‘superpathway of cholesterol biosynthesis’, ‘BAG2 signaling pathway’, ‘Wnt/β-catenin signaling’ were the topmost upregulated while ‘senescence pathway’, ‘ATM signaling’, ‘role of BRCA1 in DNA damage response’, ‘role of CHK proteins in cell cycle checkpoint control’ and ‘p53 signaling’ were top five downregulated pathways (Appendix A). We then evaluated the oppositely regulated pathways, which may drive different migration phenotypes observed in directional migration and radial migration. As expected, ‘HIF1α signaling’ is significantly upregulated in directional migration and downregulated in radial migration (Figure 2D). In hypoxia-induced directional migration, ‘role of CHK proteins in cell cycle checkpoint control’, ‘p53 signaling’, ‘senescence pathway’ and ‘sirtuin signaling pathway’ were significantly upregulated while the ‘sphingosine-1-phosphate signaling’, ‘estrogen receptor signaling’ and ‘superpathway of cholesterol biosynthesis’ were downregulated. In the secretome-induced radial migration, these pathways were oppositely regulated in addition to the significant upregulation of ‘interferon signaling’ and downregulation of ‘T cell exhaustion signaling pathway’ and ‘p38 MAPK signaling’.

### 3.3. Hypoxia Is Enriched Only in Large Microtumors with Directional Migration

To examine the influence of hypoxia in the directional and radial migration observed in our two microtumor models (600D6 and 150CM, respectively), we used Correlation Engine (CE) to obtain significance of correlation of our gene expression data with gene ontology (GO) term ‘response to hypoxia’ (Figure 3A,B) and GSEA to obtain enrichment score for ‘hallmark hypoxia’ gene set (Figure 3C). The rationale for using both GSEA and CE is the difference in the statistical algorithm used by each software to perform the enrichment. GSEA implements a running-sum statistic to calculate the enrichment score (ES) and uses empirical phenotype-based permutations to estimate significance [35]. CE uses a running fisher algorithm analogous to GSEA that calculates a correlation score. However, the statistical significance is estimated by a Fisher’s exact test rather than by permutations [57]. CE results indicated that 221 genes belong to ‘response to hypoxia’ GO term, of which 33 genes (14.9%) were common with the DEGs in the directional migration, while only 8 genes (3.6%) were common with the DEGs in the radial migration (Figure 3A,B). The p-value of overlap with ‘response to hypoxia’ was 8 orders of magnitude higher in the directional migration compared to radial migration (p = 1.6 × 10^−12^ and 4 × 10^−4^, respectively) (Figure 3B). GSEA results demonstrated that ‘hallmark hypoxia’ gene set collection with 200 genes [58] is enriched only in directional migration (NES = 4.19, Figure 3C). From the results in CE and GSEA, we then extracted and compiled the genes in ‘response to hypoxia’ biological process, and hallmark ‘hypoxia’ that are common to the DEGs in each, the directional and radial migration phenotypes. These group of genes are denoted hereafter as 79-gene tumor-intrinsic hypoxia signature and 8-gene secretome-induced hypoxia signature, respectively. For directional migration, out of 79-gene tumor-intrinsic hypoxia signature, 46 genes belong exclusively to GSEA, 24 genes to CE, and 9 genes were common between GSEA and CE. In secretome-induced radial migration, the hypoxia signature consists of only 8 genes that overlapped with ‘response to hypoxia’ in CE. We speculate that adding hypoxic secretome to non-hypoxic tumors can induce gene expression changes that fall under the term ‘response to hypoxia’ despite the fact the radially migrating small tumors do not develop intrinsic hypoxia. In a further CE meta-analysis, we obtained the significance of overlap with GO term ‘response to hypoxia’ for seven publicly available genomic studies (Appendix A) that examined the gene expression changes in breast cancer tissue or cell lines in response to hypoxic conditions. We further validated our results obtained for tumor-intrinsic hypoxia, and secretome-induced hypoxia gene signatures by comparing with these publicly available datasets (Figure 3D). The *p*-value of overlap in hypoxia-induced directional migration model with the term ‘response to hypoxia’ (*p* = 1.6 × 10^−12^) was comparable with other reported studies comparing invasive tumors with ductal carcinoma in situ (GSE3893, *p* = 4.9 × 10^−12^), MCF7 cells exposed to 4 h hypoxia (GSE19123, *p* = 1.7 × 10^−11^), or 24 h hypoxia (GSE70805, *p* = 4.7 × 10^−14^), or 48 h hypoxia (GSE47533, *p* = 9.30 × 10^−16^) or 24 h hypoxia and lactate (GSE29406, *p* = 2.2 × 10^−10^) (Appendix A). On the other hand, the significance of overlap of secretome-induced radial migration with the term ‘*response to hypoxia*’ (*p* = 4 × 10^−4^) is considerably lower than that obtained for studies GSE3893, GSE19123, GSE29406, GSE70805, and GSE47533 (Appendix A). The expression profile of the genes related to ‘*response to hypoxia*’ in our hypoxia- and secretome-induced migration models was negatively correlated to the ones in studies GSE9649 and GSE30019 (Appendix A), which was expected. This is because these studies compared the genomic changes of human breast cancer cell monolayers cultured in hypoxic conditions with lactic acidosis and re-oxygenation respectively, which may cause different signaling events when compared to our study, where we analyzed hypoxic vs. non-hypoxic conditions in 3D cultures.

Figure 3E,F show heatmaps of the genes associated with the hypoxia gene signatures of each migratory phenotype, displaying a high level of expression in red and a low level of expression in blue.

There were only three genes common between the hypoxia gene signatures of the two comparisons (Figure 3G). ADM was upregulated in both groups. However, ATM and KCNMA1 were upregulated in directional migration and downregulated in radial migration. We experimentally confirmed that the directionally migrating tumors develop intrinsic hypoxia (Figure 3H(i)) while secretome-treated small radially migrating microtumors develop none to little hypoxia (Figure 3H(ii)) using Ru-dpp-based hypoxia stain [16], further validating our microarray analysis.

### 3.4. Tumor-Intrinsic Hypoxia Induces Epithelial-Mesenchymal Transition (EMT)

We then sought to delineate the contribution of epithelial mesenchymal transition (EMT) to induce the observed modes of directional and radial migration in response to tumor-intrinsic hypoxia and secretome, respectively. We obtained enrichment scores for GSEA hallmark ‘*epithelial to mesenchymal transition (EMT)*’ for DEGs in both migratory phenotypes. Hallmark ‘*EMT*’ was enriched only in the large microtumors showing tumor-intrinsic hypoxia and directional migration (NES = 2.32, Figure 4A) and was not enriched in small non-hypoxic microtumors that exhibit secretome-induced radial migration. In the directional migratory phenotype, 26 genes (13%) overlapped with the hallmark EMT gene set. These genes are hereafter denoted as tumor-intrinsic hypoxia-induced 26-gene EMT signature.

The heatmaps of the genes associated with the EMT gene signature of directional migration are shown in Figure 4B, displaying a high level of expression in red and a low level of expression in blue. We further performed the meta-analysis of hypoxia-induced 26-gene EMT signature from directional migratory phenotype with publicly available gene expression datasets similar to Section 3.3 (Appendix A, Figure 4C). The *p*-value of overlap of tumor-intrinsic hypoxia-induced EMT gene signature was highest with GSE3893 (*p* = 2.4 × 10^−27^) comparing invasive vs. ductal breast carcinoma, followed by GSE29406 (*p* = 9.52.4 × 10^−6^) and GSE47533 (*p* = 6.2 × 10^−7^) where MCF-7 cells were treated under hypoxic conditions. Positive correlation was also observed with datasets from studies GSE19123 (*p* = 0.073) and GSE70805 (*p* = 0.0373) where MCF-7 cells were grown in hypoxic conditions for 4 h and 24 h, respectively, and compared with cells cultured under normoxia. This suggests that the proposed tumor-intrinsic hypoxia-induced 26-gene EMT signature is enriched in large hypoxic microtumors with directional migration. Similar to results from Section 3.3, EMT gene signature negatively correlated with gene expression data from studies GSE9649 (*p* = 4.6 × 10^−5^), and GSE30019 (*p* = 0.0149) (Figure 4C).

### 3.5. The Process of Migration Is Enriched Equally in Both Directional and Radial Migratory Phenotypes

To determine if the process of migration is enriched in both the directional and radial migratory phenotypes, we examined the significance of correlation of the DEGs with the GO term ‘*regulation of cell migration*’ from CE and enrichment score for ‘*tissue migration*’ gene set from GSEA. Out of 375 genes that belong to the ‘*regulation of cell migration*’ GO term, 40 genes (10.6%) overlapped with DEGs in the directional migration, while 14 (3.7%) overlapped with the DEGs in the radial migration (Figure 5A). Despite a higher percentage of overlap in the directional migration compared to radial migration, the *p*-value of overlap was comparable between the two groups (4.1 × 10^−9^ and 7.0 × 10^−7^, respectively) (Figure 5B). Moreover, GSEA enrichment scores for both directional and radial migration showed similar NES for the ‘*tissue migration*’ gene set (1.724 vs. 1.721 respectively, Figure 5C). We also performed a CE meta-analysis to obtain the significance of overlap with ‘*regulation of cell migration*’ GO term of the seven publicly available genomic studies that we previously used (Appendix A). Figure 5D shows the comparison of these results with the significance of overlap obtained for directional and radial migration phenotypes with ‘*regulation of cell migration*’. The *p*-value of overlap for the directional migration with ‘*regulation of cell migration*’ (*p* = 4.10 × 10^−9^) was comparable with studies where MCF7 cells were cultured under 24 h hypoxia (GSE70805, *p* = 4.7 × 10^−8^), 48 h hypoxia (GSE47533, *p* = 2.2 × 10^−9^), and 24 h hypoxia and lactate (GSE29406, *p* = 4.6 × 10^−11^). The significance of overlap for radial migration with ‘*regulation of cell migration*’ (*p* = 3.6 × 10^−6^) was also comparable with the studies GSE29406, GSE70805, and GSE9649 (*p* = 2 × 10^−4^) (Appendix A). We note that the expression profile of the genes related to ‘*regulation of cell migration*’ in hypoxia- and secretome-induced distinct migration modes was negatively correlated to that in study GSE30019 (Appendix A)), which analyzed gene expression of MCF7 cells exposed to 24 h hypoxia vs. 12 h re-oxygenation, indicating that re-oxygenated cells downregulate genes related to cell migration.

Similar to Section 3.3 and Section 3.4, we compiled the common genes between each of our hypoxia- and secretome-induced migration and both the ‘regulation of cell migration’ GO term from CE and ‘tissue migration’ gene set from GSEA (Figure 5E). The compiled migration related genes for each group are denoted hereafter as 69-gene directional migration signature and 21-gene radial migration signature (Figure 5E,F). Out of 69-gene directional migration signature, 30 genes are derived from GSEA, 27 genes are derived from CE and 12 genes are common between GSEA and CE. The secretome-induced radial migration gene signature consists of 21 genes, of which 7 genes are extracted from GSEA, 11 genes are from CE, and 3 genes are common between GSEA and CE. We identified only 6 genes that were common to both directional and radial migration signature gene sets (Figure 5G). Genes EFNA1, KLF4, and JUN were upregulated while TGFBR1 was downregulated in both directional and radial migration. On the other hand, AKT2 and IFITM1 were downregulated in directional migration and upregulated in radial migration.

### 3.6. Directional and Radial Migration Modes Emerge from Different Molecular Drivers with Distinct PPIs That Participate in the Same Migration-Related Pathways

We then set out to understand how directional and radial migration modes are driven by tumor-intrinsic hypoxia and secretome. We first analyzed 79-gene tumor-intrinsic hypoxia, 26-gene EMT, and 69-gene directional migration signatures to establish how tumor-intrinsic hypoxia drives directional migration through EMT. The functional classification of biological processes coding for tumor-intrinsic hypoxia, EMT, and migration signature sets in both, the directional and radial migrating microtumors is presented in Appendix A. In tumor-intrinsic hypoxia-induced migration, out of the hypoxia (Figure 3E), EMT (Figure 4C), and directional migration gene signatures (Figure 5E), eight genes were common between hypoxia and EMT gene signatures (TNFAIP3, VEGFA, JUN, LOX, TGFBI, PLOD2, CXCL12, THBS1), 12 genes were common between hypoxia and migration signatures (VEGFA, JUN, NOS3, EFNA1, PRKCA, SIRT1, THBS1, CITED2, CXCL12, SMAD3, CXCR4 and ICAM1), and eight genes were common between EMT and migration gene signatures (VEGFA, JUN, PFN2, ECM1, LRP1, TIMP1, CXCL12, and THBS1). Of the hypoxia, EMT, and migration gene signatures, four genes (VEGFA, JUN, CXCL12, THBS1) were common among all three signatures (Figure 6A). Similarly, in secretome-induced migration, two genes were common between the hypoxia and migration gene signatures (LMNA and TGFBR3) (Figure 6D).

We then constructed the PPI networks combining the hypoxia, EMT, and migration signatures from directional migration (Figure 6B) and between hypoxia and migration signatures in radial migration (Figure 6E). The directionally migrating tumors displayed a zero-order network consisting of seed genes that have direct interaction without intermediate nodes (Figure 6B). This network of directly interacting genes consisted of a total of 36 genes, of which 15 genes belonged to the 79-gene tumor-intrinsic hypoxia signature, 2 genes belonged to the 26-gene hypoxia-induced EMT signature, and 8 genes belonged to the 69-gene directional migration signature. The PPI network for directional migration also displayed 7 genes common between hypoxia and directional migration signatures, and 4 genes were common between hypoxia, EMT, and directional migration signatures (VEGFA, JUN, CXCL12, THBS1). In case of radial migration, the minimum interaction network consisted of 47 genes, of which 6 genes belonged to the secretome-induced hypoxia signature and 19 genes belonged to the radial migration signature. The PPI network for radial migration also displayed 2 genes (TGFBR3, LMNA) common between hypoxia and radial migration, and 16 genes were intermediate genes added to connect the seed genes (Figure 6E).

These results emphasize that there is no direct protein-protein interaction (PPI) between the hypoxia and migration signature genes in the radial migratory phenotype, further confirming that this phenotype may not be directly driven by hypoxia. We then obtained the clustering coefficient and network heterogeneity parameters to evaluate the connectivity of each network using the NetworkAnalyzer tool from Cytoscape [59]. The PPI between the tumor-intrinsic hypoxia and directional migration signature genes showed higher clustering coefficient (0.13) than radial migration signature (0.013), and higher network heterogeneity (0.8 vs. 0.08, respectively) indicating higher connectivity between the nodes in the network and overall influence of the proteins in the network. KEGG pathway analysis was performed on the minimum and first-order interaction networks (Figure 6B, Supplementary Data S1) and revealed that the pathways enriched in PPIs in both the directional and radial migration were similar even though the networks were built from different genes (Figure 6C,F). From both the minimum and first order interaction networks, we identified migration-related pathways such as FOXO signaling pathway, PI3K-AKT signaling pathway, and focal adhesions, in both groups. Surprisingly, the hypoxia-related HIF-1 signaling pathway was enriched in both groups. However, HIF-1 signaling was one of the more enriched pathways in minimum network of directional migration with a higher enrichment *p*-value (*p* = 1.71 × 10^−19^ vs. 3.5 × 10^−3^) and higher number of genes (25 vs. 2) compared to radial migration. We also identified the genes with high connectivity and with more participation in the enriched pathways (Figure 6C,F, Appendix A, Supplementary Data S1). In directional migration, MAPK1, MAPK3, AKT1, AKT2, PIK3R2, PIK3R3, EGFR, VEGFA, IKBKB, PRKCA, GRB2, MAPK8, KRAS, and PDGFRB (cut off ≥ 5) showed the highest connectivity and most participation in the KEGG pathways (Figure 6C, Appendix A). In case of radial migration, JUN, ATM, NR4A1, and TGFBR1 are the genes with more connectivity while JUN, AKT2, and TGFBR1 showed more participation in the enriched pathways (Figure 6F, Appendix A, Appendix A).

### 3.7. Drivers of Directional and Radial Migration Are Associated with Poor Patient Survival

To evaluate whether the compiled signature profile showed any potential for survival prognosis, we analyzed the Kaplan survival plots of genes in the tumor-intrinsic hypoxia, hypoxia-induced EMT, directional migration, and radial migration signatures (Figure 7). Five genes from the tumor-intrinsic hypoxia signature set (CXCR4, FOXO3, LDHA, NDRG1, NOS3), two genes each from the hypoxia-induced EMT (EFEMP2 and MGP), and 2 genes from directional migration signatures (MAP3K3 and PIK3R3) were associated with poor patient survival from the TCGA—breast carcinoma dataset (Figure 7A).

Indeed, histological images from Human Protein Atlas revealed overexpression of these proteins in breast carcinoma patients compared to healthy tissue. A similar analysis for the hypoxia and radial migration signature genes indicated that two genes from hypoxia signature (ATM and KCNMA1), and four genes from radial migration signatures (KLF4, EFNA1, IFITM1, and TGFBR1) were associated with poor patient survival (Figure 7B).

## 4. Discussion

Collective cell migration is a key feature of transition of ductal carcinoma in situ (DCIS) to invasive ductal carcinoma (IDC), yet the microenvironmental factors and the underlying mechanisms that trigger collective migration remain poorly understood. It has been widely shown that growing solid tumors develop intrinsic hypoxia due to the limited diffusion of nutrients and oxygen through the solid cell mass, causing changes in signaling pathways that further lead to the emergence of aggressive migratory phenotypes [21]. Hypoxia can also lead to secreted factors that can act as extrinsic signaling, further reshaping the microenvironment and inducing phenotypic changes. In this study, we used three-dimensional (3D) microtumor models to investigate gene expression changes observed in distinct modes of collective migration in response to two different microenvironmental factors, tumor-intrinsic hypoxia, and tumor-secreted factors (secretome).

In our discrete-sized microtumor models, we can generate large (600 µm) microtumors that exhibit tumor-intrinsic hypoxia and directional migration without any external stimulus, while small (150 µm) non-hypoxic microtumors exhibit radial migration only when exposed to secretome of large hypoxic microtumors [17]. To investigate genomic differences between hypoxia- and secretome-induced directional vs. radial migration modes, we analyzed differential gene expression profiles of hypoxia- and secretome-induced migratory microtumors using non-hypoxic, non-migratory small microtumors as controls.

The statistical enrichment of large tumors with directional migratory phenotype showed that the DEGs are principally related to hypoxia and the upregulation of hypoxia-regulated downstream events, such as inflammatory response and EMT. The statistical enrichment for the small non-hypoxic tumors with secretome-induced radial migratory phenotype indicated that the DEGs are associated with response to extracellular stimulus, inflammatory response, and TNFα singling.

GSEA analysis demonstrated that ‘hallmark hypoxia’ was upregulated and was the most enriched gene set in directionally migrating microtumors. On the contrary, ‘hallmark hypoxia’ was not enriched in the radially migrating small microtumors. In CE statistical enrichment, both directional and radial migratory tumors have overlapping genes with GO biological process ‘response to hypoxia’. However, the number of genes, significance, and ratio of overlap was considerably higher in directional migratory phenotype. We conjecture that even though the radially migrating microtumors do not develop hypoxia, the treatment with the secretome from large hypoxic microtumors may contain cytokines/chemokines secreted in response to hypoxia, which in turn can induce genomic changes that fall under the GO term ‘response to hypoxia’. Our previous results show that inhibition of hypoxia in directionally migrating tumors was more effective only when HIF-1α is inhibited at early stages, and that inhibition of tumor-secreted factors such as soluble E-cadherin (sE-CAD) and matrix metalloproteinases (MMPs) at any timepoint was effective in halting tumor migration [17]. These prior results demonstrated that intrinsic hypoxia initiates the directional migratory phenotype, which is subsequently maintained by tumor-secreted factors in a feedback loop [17].

A further GSEA analysis demonstrated that ‘hallmark Epithelial to mesenchymal transition (EMT)’ was uniquely enriched in the directional migration, but not in the radial migration, suggesting that EMT may not be required in radial migration. Consistent with the bioinformatic analysis, we earlier reported the upregulation of mesenchymal markers such as vimentin (VIM) and fibronectin (FN) without loss of epithelial marker E-CAD in directionally migrating tumors, suggesting the presence of a transient/partial EMT [17]. Moreover, we found increased levels of sE-CAD, FN, and MMP9 in the secretome of directionally migrating tumors [17], indicating that intrinsic hypoxic environment triggers directional migration and induces higher levels of tumor-secreted factors through the acquisition of mesenchymal features. Supporting our prior results, our microarray analysis confirmed that treatment of tumor-secreted factors from the large, hypoxic, directional migratory tumors induce genomic changes in the non-hypoxic small tumors to initiate migration in a radial fashion. Overall, enrichment of hallmark hypoxia, hallmark EMT, and migration terms from GSEA and CE analysis in large microtumors suggests interplay between tumor-intrinsic hypoxia and EMT to induce directional migration. In contrast, only *migration* term from GSEA and CE analysis was enriched in the secretome-induced migratory phenotype, suggesting that tumor-secreted factors present in the hypoxic CM are sufficient to induce radial migration in non-hypoxic tumors and EMT may not be always required for emergence of migratory phenotypes.

From our gene expression analysis, we propose unique gene signatures that are directly related to tumor-intrinsic hypoxia (79-gene hypoxia signature), hypoxia-induced EMT (26-gene EMT signature), 69-gene directional migration signature, and 21-gene radial migration signature. PPI network analysis of the tumor-intrinsic hypoxia, hypoxia-induced EMT, and directional migratory genes revealed that these signature gene sets were highly connected in the directional migratory tumors. However, there was no direct connection or PPI between the genes associated with the hypoxia and radial migration signature genes in the secretome-induced radial migratory phenotype. Interestingly, however, KEGG enrichment analysis of the minimum and first-order interaction PPI networks revealed mostly the same pathways enriched in both, directional and radial migratory phenotypes although genes participating in each pathway are either different or if common, they are oppositely regulated (downregulated or upregulated). This further suggests that the two migratory phenotypes (directional and radial) emerge from differential regulations or crosstalk between signaling molecules, activating different downstream mechanisms.

We used a meta-analysis to compare our genomic data with seven publicly available studies that analyzed the effect of hypoxia in the gene expression of breast cancer cell lines or breast cancer tissue. We found that the enrichment of the term ‘response to hypoxia’ in the studies GSE3893, GSE19123, GSE29406, GSE70805, and GSE47533 is comparable to the enrichment in our data, indicating a similar genomic profile in terms of hypoxia signature in these studies. The enrichment of the terms ‘EMT’ and ‘regulation of cell migration’ in the study GSE29406 was comparable to the ones obtained for our data. This further suggests a similar genomic profile of our data in terms of EMT and migration markers with the ones in the study GSE29406. We have also found similarity of the results with the ones obtained by El Guerrab et al., where the authors investigated the prognostic value of hypoxia-related gene expression in breast cancer, using a comparative analysis of hypoxia markers according to clinicopathological data [60]. Similar to our study, the authors found upregulation of the hypoxia-related genes ENO1, FOXO3, VEGF, LDH, and NPRG1 in recurrent patients with high grade breast cancer tumors [60]. In the work of Nair et al., the authors built and validated individual predictors of breast cancer proliferation and migration levels from the transcriptomics of 40 breast cancer cell lines [61]. Then, the authors applied these predictors to estimate the proliferation and migration levels of more than 1000 TCGA breast cancer tumors. The results of this study demonstrated that predicted tumor migration levels are significantly more strongly associated with patient survival than the proliferation levels. This study also found migration enhancer genes such as FOSL2 and NFIL3, which are present in our signature gene sets [61]. KEGG analysis of the migration enhancer genes indicated that ‘HIF-1 signaling pathway’ is the most enriched pathway in the migration enhancer genes, which is consistent with our results (Figure 6) [61]. Chen et al. used a microfluidic migration platform and single-cell RNA sequencing to investigate gene expression profiles among three migratory breast cancer cell lines and patient-derived cells [62]. Similar to our findings, the authors revealed that migratory cells exhibited signatures of EMT and that depending on the migratory phenotype, this EMT signature can vary. However, a common denominator was the upregulation of the ‘HIF-1-alpha transcription factor network signaling pathway’. The authors found overlapping DEGs in migratory cells as compared to non-migratory cells in all three cell lines. These genes included TIMP1, HMGB1, ENO1, SDC2, and PLOD2, which are also confirmed in our signature sets. These prior results further validate our results and highlight the clinical relevance of our proposed gene signature sets.

Survival analysis from DEGs on hypoxia-induced directional migration and secretome-induced radial migration revealed that drivers of both directional and radial migration are associated with poor survival rates in IDC patients. This indicates that individual factors in the tumor microenvironment play a key role in promoting tumor aggressiveness and hence, poor clinical outcomes.

Our results highlight the phenotypic heterogeneity and plasticity of collective migration under hypoxia and secretome as two different microenvironmental triggers of migration using 3D cultures generated from the same T47D cell line. We also show association between microenvironment-related unique gene signatures observed in our 3D models and poor patient survival, demonstrating the importance of each tumor microenvironmental factor in the emergence of aggressive migratory phenotypes.

## 5. Conclusions

Using engineered 3D models of breast cancer, we can recapitulate two distinct (directional vs. radial) modes of collective migration induced by different microenvironmental factors solely defined by microtumor size (tumor-intrinsic hypoxia) and tumor secretome. To the best of our knowledge, this is a unique experimental approach that allows us to recreate microenvironmental events such hypoxia and secretome, without external manipulation of oxygen gradients or chemical factors. Bioinformatic analysis of our microarray data suggests that directional migration is related with an interplay between tumor-intrinsic hypoxia and EMT. Interestingly, our analysis also suggests that tumor-secreted factors present in the hypoxic microenvironment are sufficient to induce radial migration in non-hypoxic tumors, and EMT may not be always required for emergence of all migratory phenotypes. From statistical enrichment analysis, we obtained unique gene signatures related with each tumor-intrinsic hypoxia-induced directional migration and secretome-induced radial migration. These unique gene signatures are associated with poor clinical outcomes, demonstrating potential as a survival prognosis tool. Our work highlights the importance of tumor microenvironment in the emergence of distinct migratory phenotypes as well as the phenotypic and genotypic heterogeneity of collective migration.

## Data Availability

Microarray data is deposited in the Gene Expression Omnibus database at the National Center for Biotechnology Information (GSE166211).

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
