# Peer review of "Identifying Molecular Signatures of Distinct Modes of Collective Migration in Response to the Microenvironment Using Three-Dimensional Breast Cancer Models"

_cancers, 2021, doi:10.3390/cancers13061429_

Round 1

Reviewer 1 Report

Authors aimed to identify molecular signatures of distinct modes of collective migration in response to the microenvironment. They were using three-dimensional breast cancer models created by them, where  they compare the transcriptome of collectively and radial migrating cells induced by hypoxia or secretome of hypoxic tumors.

Authors presented interesting data and following are comments and concerns to improve this manuscript:

  • Authors described that big, hypoxic tumors migrate in a directional way. Do the authors specify / studied what is the factor which determine the direction of migration – in the fig. 2C it seems that all tumors migrate in similar direction. Is there some attractant used? Did authors analysed the features of cells migrating at the front of tumors? Do they present some molecular signatures which determine the direction of migration?
  • Authors showed that culture media (CM) from hypoxic tumors stimulate migration of small, non migrating tumors. Moreover authors compared changes in gene expression between these two types of tumors, however it would be more informative if authors also included in their studies the secretome analysis at the protein level, to identify which specific factors present in CM of hypoxic tumors “force” the non-migrating ones to move.
  • The discussion part should be modified, because in the present form it is rather description and repetition of the results without referring to the literature. In this section authors cite only two articles, which were published at least partially by the same authors as the submitted manuscript.

Author Response

Response to Reviewer’s 1 comments

Point 1: Authors described that big, hypoxic tumors migrate in a directional way. Do the authors specify / studied what is the factor which determine the direction of migration – in the fig. 2C it seems that all tumors migrate in similar direction. Is there some attractant used? Did authors analyzed the features of cells migrating at the front of tumors? Do they present some molecular signatures which determine the direction of migration?

Response 1: We thank the Reviewer for this observation. In our system, we don’t use any attractant or create oxygen or chemical gradients to promote migration in a certain direction. The tumors grow in a conventional humidified incubator with 5% CO2. We also use regular culture media that is replaced every day. In the 1x1 cm2 device with an 8x8 array of 600µm wells, we consistently observe tumors migrating in different directions. In figure 2C of the original manuscript, we are showing an area of the hydrogel device with four migrating tumors that happen to be migrating in the same direction. We acknowledge that this can cause confusion. Therefore, we have replaced that image with the one that shows tumors migrating in different directions (Fig 1C of the revised manuscript).

Interestingly, we have observed that the migratory front of the large migrating tumors is smooth and ruffly. In contrast, in the small radially migrating tumors, we observe multiple migratory fronts with the appearance of invasive-like strands. Studies are currently underway to identify the molecular signatures of cells migrating at the front of tumors.

Point 2: Authors showed that culture media (CM) from hypoxic tumors stimulate migration of small, non-migrating tumors. Moreover, authors compared changes in gene expression between these two types of tumors, however it would be more informative if authors also included in their studies the secretome analysis at the protein level, to identify which specific factors present in CM of hypoxic tumors “force” the non-migrating ones to move.

Response 2: We acknowledge that CM of the microtumors contains multiple factors, and that it is crucial to understand what specific proteins are leading to the migratory phenotype. Although a full secretome analysis is currently undergoing, our group has studied soluble E-Cadherin (sE-CAD), Fibronectin (Fib), and Matrixmetalloproteinase-9 (MMP9) in the CM of 600 µm and 150 µm microtumors [1]. The CM of the large hypoxic microtumors showed increased levels of sE-CAD, FN, Pro-MMP9, and MMP9 compared with the small ones. We include this statement in the introduction of the revised manuscript (Lines 117-121)

We have also shown that the treatment with recombinant human sE-CAD (Rec-sE-CAD) alone also induced migration in the 150 µm microtumors similar to the treatment with the CM of 600 µm microtumors. These results confirmed that either the CM from 600 or sE-CAD alone could induce the migratory phenotype in nonmigratory 150 µm microtumors. Moreover, our prior studies demonstrated that the secretome from the small non-migratory 150 microtumors did not affect the migratory behavior of either small or large microtumors [1].

Point 3: The discussion part should be modified, because in the present form it is rather description and repetition of the results without referring to the literature. In this section authors cite only two articles, which were published at least partially by the same authors as the submitted manuscript.

Response 3: We thank the Reviewer for this valuable suggestion. We agree that the discussion needs to be expanded. In the discussion section of the revised manuscript, we are now including a comparison of our results with the ones obtained by other three studies that analyzed the genomic profile of breast cancer cells (Lines 659-685).

References

  1. Singh, M.; Tian, X.-J.; Donnenberg, V.S.; Watson, A.M.; Zhang, J.; Stabile, L.P.; Watkins, S.C.; Xing, J.; Sant, S. Targeting the temporal dynamics of hypoxia-induced tumor-secreted factors halts tumor migration. Cancer research 2019, 79, 2962-2977.

Reviewer 2 Report

The authors studied to identify a key signature for cancer cell migration induced by tumor intrinsic hypoxia, and secretome-induced migratory behavior. The authors analyzed differential gene expression to identify the main genetic signatures in these bioprecess. The authors found that 1992 genes (734: up, 1258: down) for directionally migrating microtumor, and 305 genes (189: up, 116: down) for radially migrating microtumors. Furthermore, the authors found identified 36 hallmark genes that are involved in the regulation of hypoxia and G2M checkpoint regulation in hypoxia-induced directional migration, and 2 hallmark genes that are involved in the regulation of TNFa signaling via NF-kB, and mTORC1 signaling in secretome-induced radial migration. The analysis was expanded to identify EMT related signature, migration signature, PPI networking as well as patient survival. In summary, the authors obtained big and invaluable data, which has great potential to mining gold. Question is that the omics study is not proved yet. The reviewer doesn’t know the real role of each genes in biological process whether each gene in particular signature plays a key role in the suggested process. Are there any opposite publications? If the authors may not say yes, the authors should provide the several evidence, at least, that the signatures and models are reflected the physio-pathological relevance. If not, the results are a well-organized result with very high potential to mimic pathophysiological relevance, but not first-class cooking.

Author Response

We acknowledge the Reviewer’s concern. However, we would like to emphasize that the gene signature sets proposed in this manuscript derived from a statistical enrichment of genomics data in our 3D microtumor models using already compiled and published gene sets from the Molecular Signatures Database (MSigDB) collections. Statistical enrichment of our data against those collections demonstrated that Hallmark hypoxia, Hallmark EMT and Tissue migration were enriched in the genomic data obtained from our 3D models. Given that our signature sets come from this already curated database of the functions we are studying, we believe that our work demonstrates that each gene in the signature set plays a key role in the suggested process, namely hypoxia, EMT or migration. We believe that our work demonstrates the pathophysiological relevance of the proposed gene sets based on the statistical enrichment and the survival analysis.

Reviewer 3 Report

The submitted manuscript is a research article focused on profiling of differentially expressed genes (DEGs) in breast cancer cell line model T47D. The authors used two size-controlled microtumor models, small (˃150µm) and large (˃500µm), which exhibit distinct collective migration modes, directional and radial, respectively, depending on the microenvironment. They showed that breast cancer cells migrate directionally in the large microtumors, and this is associated with DEG signatures under “intrinsic hypoxia” micro-environment and hypoxia-induced epithelial-to-mesenchymal transition (EMT). While cancer cells from small microtumor models did not suffer from hypoxia and demonstrated “secretome-induced” radial migration. Additionally, they identified unique genes associated with low survival rate and poor prognosis in TCGA-breast invasive carcinoma dataset.

However, there are some concerns and recommendations, which as follows:

  • The authors operate with the term “collective migration” throughout the manuscript. and use it in the manuscript title. However, the authors did not study and did not characterize features of collective migration such as coordinated movement, cell-to-cell junctions, the formation of structural and functional units with common properties (collective polarization, etc.). Thus, the use of the word “collective” is doubtful and should be excluded.
  • The authors operate with the term “intrinsic hypoxia”. This term should be explained and discussed in the Introduction, regarding the difference between “intrinsic hypoxia” and “external hypoxia”.
  • The authors operate with the term “secretome”. However, the secretome includes a wide range of factors (include growth factors, chemokines, cytokines, adhesion molecules,) secreted by cells in extracellular space. Furthermore, each cell or tissue type secretes specific molecules, and the authors should at least discuss in the Introduction and Discussion what kind of proteins are secreted by breast cancer cells. This could be made using previous publications of the authors [15]. Additionally, how can the secretome of small microtumors differ from the secretome of large microtumors?
  • Results: more attention should be given to the interpretations of biological processes derived from GO. In sections 3.2. and 3.5 and Figs. 3 and 5: GSEA GO terms contain similar terms such as “Peptide secretion” and “Regulation of peptide secretion”, “Cellular response to toxic substances” and “Response to toxic substances”, “Cellular response to light stimulus” and “Response to light stimulus”, etc. This should be addressed in more detail because the term “cellular response” is a subset of “response” and can not be directly used for gene annotation. The same is for the term “cell activation”, “tissue migration’. The term “apoptosis” should be clarified indicating “anti-“ and “pro-apoptotic”, etc.
  • Results, section 3.6: functional classification of biological processes would be useful for a more full understanding of DEG annotations. Especially, this concerns upregulated and downregulated genes.
  • The is some confusion in the indication of large microtumor size: ˃500µm in the Abstract and 600µm in the text.
  • Gene expression profiles known for breast cancer and obtained by other investigators should be discussed in Introduction and Discussion. The main gene expression profiling data can be emphasized in Conclusions.
  • Style and grammar: lines 7-13: author affiliations are not full, line 171, “1% agarose electrophoresis”; line 175: 2.4 µg, etc.

Author Response

Response to Reviewer’s 3 comments

Point 1: The authors operate with the term “collective migration” throughout the manuscript. and use it in the manuscript title. However, the authors did not study and did not characterize features of collective migration such as coordinated movement, cell-to-cell junctions, the formation of structural and functional units with common properties (collective polarization, etc.). Thus, the use of the word “collective” is doubtful and should be excluded.

Response 1: We would like to point that our earlier published work has demonstrated that large directional migrating tumors in our hydrogel microarrays move as a whole toward the wall of the microwells, and eventually the tumor moves as a unit out of the well [1]. We have also demonstrated that these microtumors conserve the cell-cell junction marker E-Cadherin during the entire migratory process [1], which is a key characteristic of collective migration [2,3] (see the images showing E-cadherin staining in non-migratory 150, migratory 150CM and 600 µm microtumors). To make this further clear, we have referred to our previous work in the introduction (Lines 121-123) to demonstrate the collective nature of the tumor migration observed in our system. Also, as per the suggestion, we have removed the word “collective” in many places and toned it down throughout the manuscript.

Point 2: The authors operate with the term “intrinsic hypoxia”. This term should be explained and discussed in the Introduction, regarding the difference between “intrinsic hypoxia” and “external hypoxia”.

Response 2: We thank the Reviewer for the suggestion. We now address this in the Introduction of the revised manuscript (Lines 84-87). We also modified another paragraph in the Introduction (Lines 97-105) that now includes an explanation of external hypoxia to highlight the difference with tumor-intrinsic hypoxia.

Point 3: The authors operate with the term “secretome”. However, the secretome includes a wide range of factors (include growth factors, chemokines, cytokines, adhesion molecules,) secreted by cells in extracellular space. Furthermore, each cell or tissue type secretes specific molecules, and the authors should at least discuss in the Introduction and Discussion what kind of proteins are secreted by breast cancer cells. This could be made using previous publications of the authors [15]. Additionally, how can the secretome of small microtumors differ from the secretome of large microtumors?

Response 3: We acknowledge that CM of the microtumors contains multiple factors, and that it is crucial to understand what specific proteins are leading to the migratory phenotype. Although a full secretome analysis is currently undergoing, our group has studied soluble E-Cadherin (sE-CAD), Fibronectin (Fib), and Matrixmetalloproteinase-9 (MMP9) in the CM of 600 µm and 150 µm microtumors [1]. The CM of the large hypoxic microtumors showed increased levels of sE-CAD, FN, Pro-MMP9, and MMP9 compared with the small ones. We include this statement in the introduction of the revised manuscript (Lines 117-121)

We have also shown that the treatment with recombinant human sE-CAD (Rec-sE-CAD) alone also induced migration in the 150 150 µm microtumors similar to the treatment with the CM of 600 150 µm microtumors. These results confirmed that either the CM from 600 or sE-CAD alone could induce the migratory phenotype in nonmigratory 150 µm microtumors. Moreover, our prior studies demonstrated that the secretome from the small non-migratory 150 microtumors did not affect the migratory behavior of either small or large microtumors [1].

Point 4: Results: more attention should be given to the interpretations of biological processes derived from GO. In sections 3.2. and 3.5 and Figs. 3 and 5: GSEA GO terms contain similar terms such as “Peptide secretion” and “Regulation of peptide secretion”, “Cellular response to toxic substances” and “Response to toxic substances”, “Cellular response to light stimulus” and “Response to light stimulus”, etc. This should be addressed in more detail because the term “cellular response” is a subset of “response” and cannot be directly used for gene annotation. The same is for the term “cell activation”, “tissue migration’. The term “apoptosis” should be clarified indicating “anti-“ and “pro-apoptotic”, etc.

Response 4: We thank the reviewer for the oversight on interpretation of GO: Biological processes. We have updated Fig. 2 of the revised manuscript (Fig. 3 of the original manuscript) and the corresponding section 3.2. with the more specific GO: ‘child terms’ which are better representative of gene annotation. (Parent term is a broader GO term, and the child term is a more specific term; http://geneontology.org/docs/ontology-relations/#:~:text=Conventions%20used%20to%20describe%20relations&text=A%20node%20refers%20to%20a,be%20a%20more%20specific%20term).

Fig 2B (i-ii)/ Section 3.2: In Fig 2B (i) of the revised manuscript, we have removed the ‘parent terms’: ‘Peptide secretion’, ‘Cell junction organization’, ‘Response to toxic substance’ and ‘Biological adhesion’. In Fig 2B (ii) we have removed the ‘parent terms’: ‘Response to light stimulus’, ‘Fibroblast proliferation’, ‘Regulation of cytoskeleton organization’, and ‘Supramolecular fiber organization’. The corresponding section 3.2.2 has been updated with revised results (Lines 336-340).

Fig 3C (i)/Section 3.5: In Fig 2C (i) of the revised manuscript presents enrichment of ‘Hallmark gene sets’ in directional migration group. ‘Apoptosis’ is one of the hallmarks, which was enriched in directional migration group. However, molecular signature database (MSigDB), which presents 50 hallmark gene sets, does not classify hallmark ‘Apoptosis’ into ‘anti-’ and ‘pro-apoptotic’ (http://www.gsea-msigdb.org/gsea/msigdb/genesets.jsp?collection=H).

Fig 5: We went through Fig 4 of the revised manuscript (Figure 5 of the original manuscript) and could not find representation of GO: biological processes. Fig 4 presents 26-gene epithelial to mesenchymal transition (EMT) gene signature set.

Point 5: Results, section 3.6: functional classification of biological processes would be useful for a more full understanding of DEG annotations. Especially, this concerns upregulated and downregulated genes.

Response 5: As suggested by the reviewer. We have performed functional classification of biological processes for hypoxia, EMT and migratory signature. We have updated the comprehensive results for functional classification of biological processes as Supplementary Table II.

Point 6: There is some confusion in the indication of large microtumor size: ˃500µm in the Abstract and 600µm in the text.

Response 6: We thank the reviewer for pointing this out. We now refer to the large microtumors as 600 µm throughout the manuscript to avoid any confusion.

Point 7: Gene expression profiles known for breast cancer and obtained by other investigators should be discussed in Introduction and Discussion. The main gene expression profiling data can be emphasized in Conclusions.

Response 7: We agree with the reviewer. In our work, we performed a meta-analysis to compare our genomic data with the one obtained by publicly available genomic studies (GSE3893, GSE19123, GSE29406, GSE70805, GSE47533, GSE9649, GSE30019). These seven studies analyzed the effect of hypoxia in the gene expression of breast cancer cell lines or breast cancer tissue. We utilized the correlation engine meta-analysis tool to obtain the significance of correlation of our data and the seven studies with “response to hypoxia’, “epithelial to mesenchymal transition”, and “regulation of cell migration”.  The results of the meta-analysis can be found in Fig. 3D, Fig. 4C, Fig. 5D, section 3.3 Lines 397-411, section 3.4 Lines 446-456, and section 3.5 Lines 476-L487. We now include a discussion of these results together with a comparison with other three genomic profiles obtained by other investigators (Lines 659-685).

Point 8: Style and grammar: lines 7-13: author affiliations are not full, line 171, “1% agarose electrophoresis”; line 175: 2.4 µg, etc.

Response 8:  We apologize for the oversight in style and grammatical errors, which are now corrected in the revised manuscript:

  • L6-L10: The authors affiliations are now full.
  • We corrected Lines 176 to “1% agarose gel electrophoresis.”
  • We corrected Lines 180 to “2.4 µg.”
  • Additionally, we checked the entire manuscript for grammar and style.

References

  1. Singh, M.; Tian, X.-J.; Donnenberg, V.S.; Watson, A.M.; Zhang, J.; Stabile, L.P.; Watkins, S.C.; Xing, J.; Sant, S. Targeting the temporal dynamics of hypoxia-induced tumor-secreted factors halts tumor migration. Cancer research 2019, 79, 2962-2977.
  2. Cheung, K.J.; Gabrielson, E.; Werb, Z.; Ewald, A.J. Collective invasion in breast cancer requires a conserved basal epithelial program. Cell 2013, 155, 1639-1651.
  3. Na, T.-Y.; Schecterson, L.; Mendonsa, A.M.; Gumbiner, B.M. The functional activity of e-cadherin controls tumor cell metastasis at multiple steps. Proceedings of the National Academy of Sciences 2020, 117, 5931-5937.

Round 2

Reviewer 2 Report

The manuscript has been improved and have been provided the explanation for the concerns.

No more question.